# FEATURE-DRIVEN TALKING FACE GENERATION WITH STYLEGAN2

## ABSTRACT

In this work, we wish to use a face image that generate a more natural and real face talking animation video. This is not an easy task because face appearance variation and semantics of speech are coupled together when tacking face have a micro movement. Audio features sometimes contain information about expressions, but they are not accurate enough. So a single audio feature cannot fully represent the movement of the face. For the above reason, we want to use different features to generate talking faces. The StyleGan series show good performance in the direction of image processing, and can perform the style migration task of portraits very well at the same time. We find that StyleGan can be used as a talking face generator. At the same time, we also encode and extract non-identity features and non-lip features, and try to find the subtle relationship between the features and the talking face. We also use the evaluation and ablation study to measure the quality of the generated videos and examine whether our approach is effective and feasible.

## 1 INTRODUCTION

Using the characteristics of multimedia to realize the interaction between virtual characters and users is one of the applications of AI technology. Audio is often easier to obtain compared with video. The task of using audio and an image of a face to generate a video animation has recently been carried out by more and more researchers to realize and explore. Solving the task is essential to achieve a wide range of practical applications, such as virtual character interaction, copying videos in other languages, video of a conference or role-playing game, and so on.

Graphics-based face animation generation methods often require a completely original video sequence as input (Liu & Ostermann, 2011) (Garrido et al., 2015) (Suwajanakorn et al., 2017). Fried et al. (2019)proposed a new method to edit the conversation header video based on its transcript to produce a real output video. However, a retimed background video is required as input in their method. It takes about 1 hour of video to produce the best quality results. There is also a model that takes into account the movement of the face and uses landmarks to drive it (Wang et al., 2019) (Zakharov et al., 2019) (Gu et al., 2020). There have also been many ways to generate facial animations through audio drivers in recent times (Jamaludin et al., 2019). Prajwal et al. (2020) use the pretrained lip-sync model and add it to the system that generates facial animation to obtain the effect of lip synchronization.

In human-to-human communication, speech sounds inevitably involve lip movements. That is, the speaker's lip movements and speech are closely related. In speech recognition and speaker recognition, the most commonly used speech features are Mel-scale Frequency Cepstral Coefficients (MFCC). Utilize MFCC to obtain speech-related features and find the relationship with the features of lip movement when a person speaks. It is possible using speech to generate mouth animation. However, it is not possible to obtain all the information about face motion with audio alone. To make the face more natural, additional features are needed to make the generated face have more realistic features.

There are many ways to generate talking faces driven by speech, we propose a method using Style-Gan2 to generate animation. At the same time, considering non-lip-related features, we try to extract some features except identity features and lip features. Combine audio features with facial features to hopefully get a more realistic facial animation.

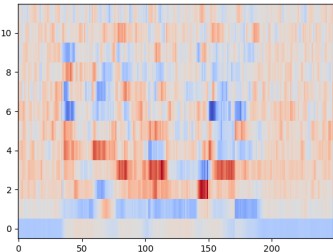 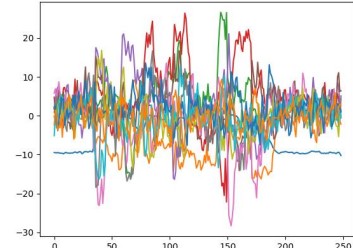

Figure 1: The left parameter dimension map and the right parameter amplitude map after a piece of audio passes through MFCC.

The rest of the paper is organized as follows: we survey the recent developments in this field in Section 2. In Section 3, we introduce our approach. In Section 4, training details are shown. After these, we evaluate our method. Finally, conclude our work in section 6.

## 2 RELATED WORK

In this section, we briefly review some related works on talking face generation.

### 2.1 GENERATIVE ADVERSARIAL NETWORK (GAN)

Generative adversarial network (GAN) is a framework that is composited of a generator network and a discriminator network. It can be utilized to train a generative model. The generator tries to fool the discriminator maximally. On the other hand, the discriminator also tries to discriminate the generated samples from the true ones as much as possible. By using this manner, both the generator and discriminator's performance can be improved in the end. GAN methods have wildly utilized for many computer vision tasks, for example, image synthesis (Radford et al., 2016), image super-resolution (Ledig et al., 2017), and image style transfer (Zhu et al., 2017). In recent years, some methods proposed to improve the original GAN from different perspectives. Such as Conditional GAN (CGAN) (Mirza & Osindero, 2014), the InfoGAN (Chen et al., 2016), and the CycleGAN (Zhu et al., 2017). The GAN methods also can be utilized in the data enhancement. For example, we can use GAN to generate different action images of people to train an action recognition model.

### 2.2 TALKING FACE GENERATION

Synthesizing high-fidelity audio-driven facial video sequences is an important and challenging problem in many applications like digital humans, chatting robots, and virtual video conferences.

For a long time, the topic of synthesizing realistic speech video from audio with an image as input is very charming for researchers. In the beginning, people adopted the model-based approach. These methods need to establish the relationship between audio semantics and lip movement. Such as phoneme mapping (Fisher, 1968)and anatomical actions (Edwards et al., 2016). Because the establishment of this relationship is quite difficult, it is not suitable for large-scale use.

In recent years, with the development of GAN technology, the research of constrained talking face generation from speech began to flourish. Kumar et al. (2017) attempted to generate key points synchronized with audio by using delay LSTM (Graves & Schmidhuber, 2005). They learn a mapping between the input audio and the corresponding lip landmarks of Barack Obama. Suwajanakorn et al. (2017) also proposed a method named "teeth proxy" for improving the visual quality of teeth generation. But these methods can only train specific people who have a lot of video data. Fried et al. (2019) proposed that the video of a single speaker can be edited seamlessly by adding or deleting phrases in the speech. Unfortunately, to accomplish this task, they still need at least one hour's data for each speaker.

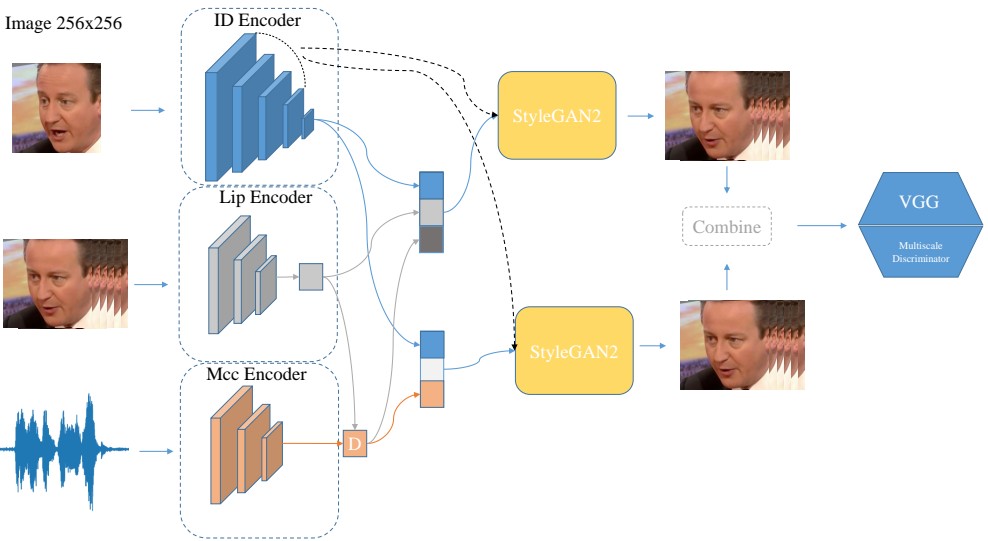

Figure 2: Schematic diagram of the end-to-end generative network model.

Subsequently, Chung et al. (2017)tried to use the encoder-decoder CNN model to learn the corresponding relationship between the original audio and video. Combined with RNN and GAN (Goodfellow et al., 2014), Jalalifar et al. (2018) used the LSTM network to create lip markers in audio input and conditional GAN (CGAN) to generate the result face image based on a specific set of lip markers. Compared with CGAN, Vougioukas et al. (2018) proposed a temporary GAN (Saito et al., 2017) to improve the synthesis quality. However, the above method is still only suitable for human faces in synthetic dataset.

Recently, people pay more attention to the synthesis technology of face for any person's conversation. Chen et al. (2018) considered the correlation between speech and lip movement when generating multiple lip images. Researchers use optical flow to better express information between frames. The optical flow not only represents the current shape information, but also represents the previous time information.

Front photos usually have identity and language information. Assuming this, Zhou et al. (2019) used an antagonistic learning method to separate different types of information from an image in the process of image generation. This kind of non-entangled representation is convenient, that is, audio and video can be used as the source of voice information. Therefore, when applying the generated network, we can not only output features, but also express them more clearly.

To find the high-level correlation between audio and video, Zheng et al. (2018) proposed a mutual information approximation to approximate the mutual information between modes. Chen et al. (2019) applied landmarks and action attention to generating talking faces. The author further proposes a temporal consistency of dynamic pixel loss. Wiles et al. Wiles et al. (2018) proposed a self-monitoring framework called x2face for learning embedded features and generating target facial motion. As long as the embedded features are learned, it can generate video from any input.

## 3 APPROACH

We propose an end-to-end architecture for taking face synthesis as shown in figure3. The generative network is an encoder-decoder structure. The system consists of generator using StyleGan2 and discriminator which evaluates the generated sequence. We adopted a modified Zhou et al. (2019) as the backbone.

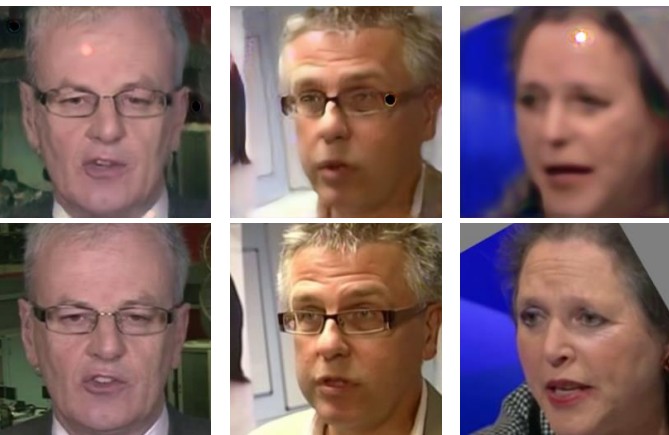

Figure 3: Images generated using upsampling have "small holes".

## 3.1 IDENTITY ENCODER

We use Identity Encoder $\mathbf{E}_i$ to get the identity information from the still frame. A U-net architecture is used with skip connections between the Identity encoder and the Decoder. The features at the five scales in the downsampling process are used as the output of the Encoder $\mathbf{E}_i$ .

## 3.2 VIDEO ENCODER

When encoding the video sequence, We want to be able to obtain lip motion features in the face, blink features of eyes, and motion features of facial contours. We modified FAN (Bulat & Tzimiropoulos, 2017) to get features. The resulting lip features are used to establish relationships with audio features, and the eye and contour features are used to influence facial movements.

## 3.3 AUDIO ENCODER

The Mel-scale frequency cepstral coefficients of the audio are used as input, and the features in the audio are obtained using $\mathbf{E}_a$.

## 3.4 DECODER

We first tried upsampling (Zhou et al., 2019) as our generator. However, when evaluating the generated results, the situation as shown in the figure appears. There will be watermark-like features such as "small holes" in the image. The StyleGan series perform well in the direction of image processing and can perform the style migration task of portraits very well at the same time. Using StyleGan2 can solve the problem of "small holes". We spliced together the audio feature, the top identity feature, and the face feature, as the styles of StyleGAN2.Take the ID encoder multi-scale features from small to large as the identity style. Also inside the generator, we spliced the features of the up-sampling part of the generator and the features of the same scale in the down-sampling stage of the identity feature together.

## 3.5 DISCRIMINATOR

A classification discriminator is used to extract the speech features in the ID encoder and make them misclassified, thus removing the speech-related information. A discriminator is also used to optimize the sequence. The generated and ground truth images are sent to a multiscale discriminator $\mathbf{D}$ with $\boldsymbol{N}_D$ layers.

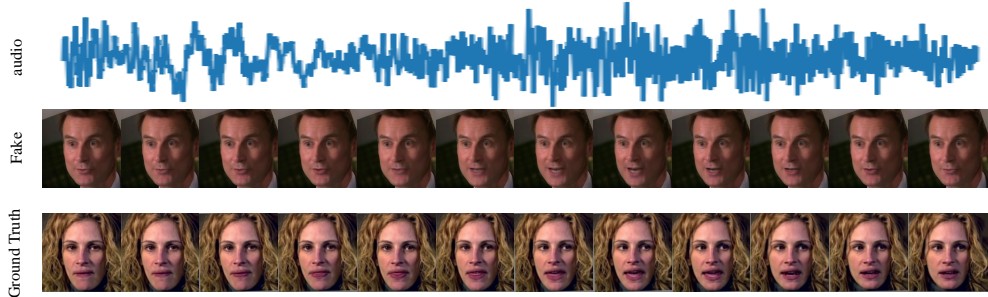

Figure 4: Audio-driven talking face generation

## 4 TRAINING

### 4.1 LIP SYNCHRONIZATION

Regarding the synchronization of the mouth and audio when performing Video Encoder, extract the mouth features of the talking face. The normalized $L_2$ loss is used to calculate the contrast loss between the audio feature and the lip feature, and an unnormalized $L_2$ contrast loss between the audio feature and the lip feature is added on this basis. At the same time, a discriminator $\mathbf{D}_{sync}$ is added for the lip feature consistency adversarial of video and speech, and $\mathcal{L}_{gan}$ is used as the loss.

$$\mathcal{L}_{sync} = \min_{\mathbf{E}_v} \max_{\mathbf{D}_{sync}} \sum_{n=1}^{N} (\mathbb{E}_{\mathbf{F}_a}[\log \mathbf{D}_{sync}(\mathbf{F}_a)] + \mathbb{E}_{\mathbf{F}_v}[\log(1 - \mathbf{D}_{sync}(\mathbf{F}_v)])$$

### 4.2 TALKING FACE GENERATION.

Combining identity feature $\boldsymbol{f}_i$ video feature $\boldsymbol{f}_v$ and audio feature $\boldsymbol{f}_a$, our system can generate a frame using the styleGan2. The newly generated frame can be expressed as $\mathbf{G}(\boldsymbol{f}_a, \boldsymbol{f}_v, \boldsymbol{f}_i)$. The generation results can be expressed as

$$\mathbf{G}(\boldsymbol{F}_a, \boldsymbol{F}_v, \boldsymbol{F}_i) = \{G(\boldsymbol{f}_a(1), \boldsymbol{f}_v(1), \boldsymbol{f}_i(k)), \dots, G(\boldsymbol{f}_a(n), \boldsymbol{f}_v(n), \boldsymbol{f}_i(k))\},$$

where $\boldsymbol{f}_i(k)$ is the identifying feature of the random $k$th frame, which acts as identity guidance.

A VGG19 reconstruction loss is used to help capture the face movement. Our overall loss function consists of a VGG19 reconstruction loss and a temporal GAN loss, where a discriminator $\mathbf{D}_{seq}$ takes the generated sequence $\mathbf{G}(\boldsymbol{F}_a, \boldsymbol{F}_v, \boldsymbol{F}_i)$ as input. These two terms can be formulated as follows:

$$\mathcal{L}_{\text{vgg}} = \sum_{n=1}^{N} ||\mathbf{VGG}(\boldsymbol{I}_{(k)}) - \mathbf{VGG}(\mathbf{G}(\boldsymbol{F}_a, \boldsymbol{F}_v, \boldsymbol{F}_i))||_1$$

$$\mathcal{L}_{\text{gan}} = \min_{\mathbf{G}} \max_{\mathbf{D}} \sum_{n=1}^{N} (\mathbb{E}_{\boldsymbol{I}_{(k)}}[\log \mathbf{D}_n(\boldsymbol{I}_{(k)})] + \mathbb{E}_{\boldsymbol{F}_a, \boldsymbol{F}_v, \boldsymbol{F}_i}[\log(1 - \mathbf{D}_n(\mathbf{G}(\boldsymbol{F}_a, \boldsymbol{F}_v, \boldsymbol{F}_i)))])$$

The overall learning objective for the whole system is formulated as follows:

$$\mathcal{L} = \lambda_{\text{gan}}\mathcal{L}_{\text{gan}} + \lambda_{\text{vgg}}\mathcal{L}_{\text{vgg}} + \lambda_{\text{sync}}\mathcal{L}_{sync}$$

Besides, the identity-preserving module of the network is trained on a subset of the MS-Celeb-1M dataset (Guo et al., 2016).

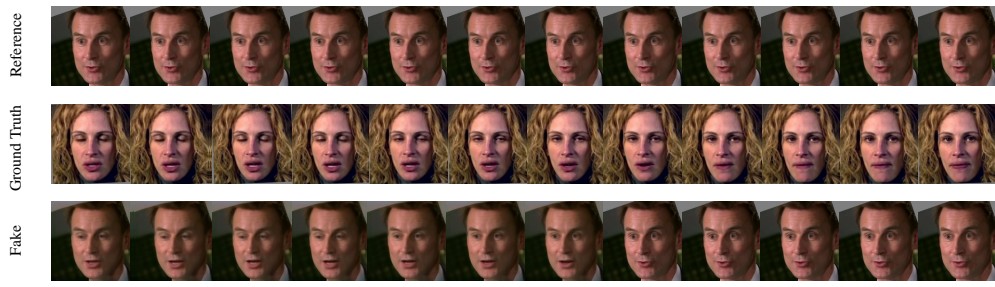

Figure 5: Video-driven talking face generation

# 5 EXPERIMENTS

Both audio and video can drive the generation of facial animations, and in order to evaluate the respective effects, our evaluation is divided into two parts.

## 5.1 DATASETS

Our model is trained and evaluated on the LRW dataset. Each sample of LRW is a 1-second video. It intercepts the head movement of the speaker when they speak a certain word. We also use the CREMA-D dataset as the test set. Each of the 91 actors speaks 12 sentences and contains the different emotions and intensities of the speakers.

## 5.2 METRICS

We use PSNR , SSIM(Wang et al., 2004), CPBD(Narvekar & Karam, 2009) as evaluation indicators to evaluate the quality of our generated videos. For the LRW dataset, audio-driven and video-driven generated videos are evaluated. CREMA-D mainly evaluates the video-driven part.See Table 1.

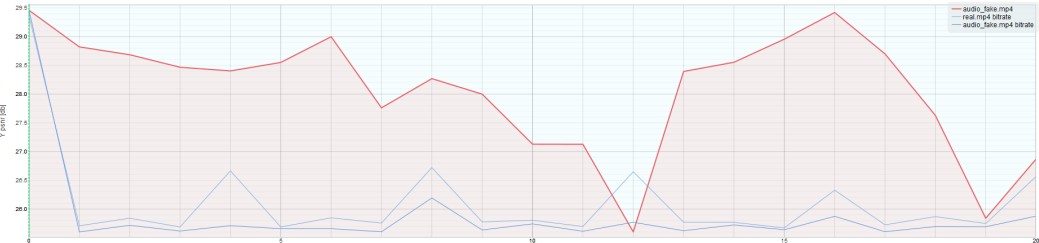

Figure 6: An audio-driven fake video and real video evaluation comparison.

Table 1: The quantitative results on LRW and CREMA-D.

| Metrics | LRW(audio) | LRW(video) | CREMA-D(video) |
|---------|-----------|-----------|----------------|
| PSNR    | 29.893    | 32.406    | 27.777         |
| SSIM    | 0.912     | 0.911     | 0.534          |
| CPBD    | 0.110     | 0.077     | 0.242          |

For the generated result, in the video-driven generated video, the eyes will blink like the ground truth, and the face will also turn like the ground truth. Audio-driven generated videos are mostly lip movements.

Table 2: Qualitative results of each method on the LRW dataset

| Method | PSNR | SSIM | CPBD |
|--------|------|------|------|
| SDA | 27.100 | 0.818 | 0.268 |
| Wav2lip | - | 0.862 | 0.152 |
| DAVS | 29.9 | 0.73 | - |
| AVTG | 20.107 | 0.810 | 0.102 |
| Ours(audio) | 29.893 | 0.912 | 0.110 |
| Ours(video) | 32.406 | 0.911 | 0.077 |

We also want to evaluate the effect of our generated video by comparing it with some existing models. Here we choose several classic talking face models for comparison, namely SDA (Vougioukas et al., 2020), Wav2lip (Prajwal et al., 2020), DAVS (Zhou et al., 2019), and AVTG (Chen et al., 2019). For SDA, visually, the actual picture is relatively blurry. Wav2lip does a better job of syncing the audio, but the facial movements are all static. DAVS has a "hole" situation. Our model outperforms these models on some metrics and falls short on others.See Table 2.

## 5.3 ABLATION EXPERIMENT

We also did some simple ablation experiments. The first is to compare the use of $L_1$ and VGG for reconstruction loss. Using VGG produces better textures and details than $L_1$.

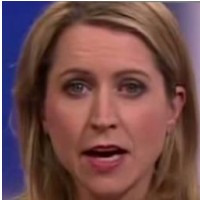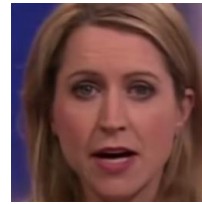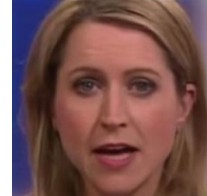

Figure 7: From left to right: ground truth, image generated with $L_1$ , image generated with VGG.

At the same time, we also remove the synchronization discriminator to see the audio-driven generated image and the movement of the mouth. A small dataset of 39 classes is used here. It can be seen that the synchronization of mouth movements is weakened.

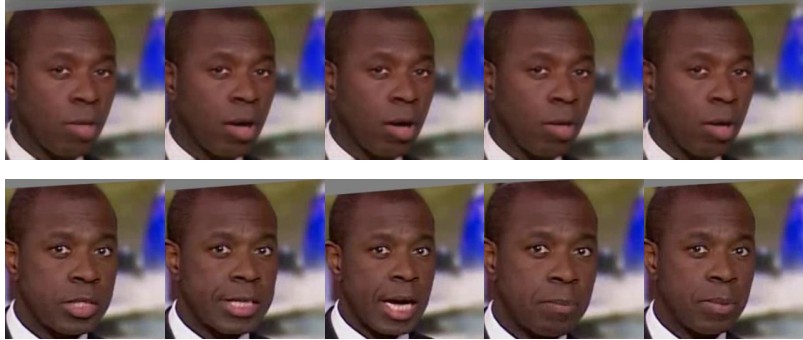

Figure 8: The effect of the synchronization discriminator: generative images and ground truth comparisons.

We implemented the method using Pytorch. The batch size is set to be 8 with 1e-4 learning rate and trained on 4 RTX3090.

## 6   CONCLUSION AND FUTURE WORK

In this work, we use an unsupervised GAN network to extract features from facial animations. Therefore, the talking face can be generated from both the voice features and the facial features. Audio-driven video can mainly generate mouth movements, while from video-driven taking face, there are not only mouth movements, but also eye and face turning movements. We evaluate the generated results, which are better than some classic models from the past. At the same time, we also show that using VGG loss can improve the quality of generated images, and the synchronous discriminator also has its role.

At present, in the video generated by the above method, when the head rotates greatly, the boundary effect is not very good. Next, we will further study the non-lip-related feature extraction method to make the generated image more realistic.

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
