# OpenReview forum: "Feature-Driven Talking Face Generation with StyleGAN2"
_ICLR.cc/2023/Conference — Submitted to ICLR 2023_

### Official Review · Reviewer_34UQ · 2022-10-24

**Confidence:** 4
**Correctness:** 1
**Technical Novelty And Significance:** 2
**Empirical Novelty And Significance:** 1
**Recommendation:** 1

**Clarity, Quality, Novelty And Reproducibility:**

For the acoustic features, what is the window size for computing the MFCCs, and what is the hop size?  I assume you use something fairly typical like a 25ms window with 10ms hop size, but you should be explicit.

Is the frame rate of the audio and video the same?  If not, how is the audio and the video aligned?

The model produces the output frame-by-frame, so how much acoustic context is provided to generate a frame of video? Is the system causal so only uses past context, or is it non-causal using both past and future context?

I am not sure what Figure 1 is trying to show.  The left figure is labelled the “parameter dimension map” — is this a visualization of the MFCCs, where the color relates to the value?  The amplitude plot in Figure 1 also does not really provide any information.

For Figure 2, you have the same person as input for the ID encoder and the lip encoder.  At test-time, these would be different individuals, so I would use different people in the example.

Expand the figure/table captions to completely describe the content independently of the main text.  Include the main takeaway for the reader.

In the approach section (Section 3), at the end of the first sentence you reference Figure 3.  This should be Figure 2.

The sub-sections in Section 3 are missing important details about the architectures.  For example, in Section 3.2 you mention modifying FAN, but there are no details about the modifications.  In Section 3.3 there are no details about the architecture for encoding the MFCCs.

In Figure 4 the caption states you are showing an audio-drive talking face, then the rows have labels “audio”, “fake”, and “ground-truth”.  What is fake?  Is this a generated face that should match the speech?  If so the lip shapes are different from the ground-truth.  If the objective is not to match the ground-truth, we have no way of evaluating the output looking at the figure because there is no information about what is being said (and so having some idea of the target lip shapes).

Explain all of the terms in your equations.

Order the rows consistently in Figures 4 and 5 — the Fake and Ground-truth are swapped.  Again, the lip shapes do not match in the Fake and Ground-truth rows.

Label the rows in Figure 8.

How are missing input modalities handled?  For example, if I wanted to generate the visual speech to align with an input audio sequence, but I do not have the accompanying video for the lip encoder?



**Strength And Weaknesses:**

+The work compares against a number of baseline approaches.

+StyleGAN models have shown to work well for multiple problems.

-The paper lacks a lot of detail that would be required to reproduce the method.  Information about the architectures, the feature processing, and so on is all missing.

-It is difficult to evaluate the quality of the output from the way the results are presented.  There are some objective measures included, but these relate more to the image quality and not the visual speech content captured in the video.


**Summary Of The Paper:**

This paper describes an approach for generating video of a talking face from a reference image using either speech as input, or using a video sequence of another person as input.  The architecture uses a separate encoder each for learning a representation of the acoustic features, the lip motion, and the identity.  These learned representations are then concatenated and used as input to StyleGAN2 models for generating the output sequence.

**Summary Of The Review:**

The approach taken in the paper seems to be a reasonable one.  Speech-driven talking faces have seen success elsewhere, as has StyleGAN2 for generating content.  However, with so many gaps in the information provided, it is impossible to gauge the quality of the work.  Likewise, the choice of objective metrics makes it difficult to understand how effective is the model at producing visual speech as the measures relate more to image quality.

---

### Official Review · Reviewer_gYGv · 2022-10-25

**Confidence:** 4
**Correctness:** 2
**Technical Novelty And Significance:** 1
**Empirical Novelty And Significance:** Not applicable
**Recommendation:** 3

**Clarity, Quality, Novelty And Reproducibility:**

The structure of this paper as well as the novelty are not clear. I could not find any concrete contribution.

**Strength And Weaknesses:**

The idea of using combine audio and image feature for generation task is interesting,

**Summary Of The Paper:**

This work is for talking animation video generation. The authors propose a generative pipeline via different feature extraction like ID, Lip and audio. Also, the authors did some experiments and ablation study for evaluation.

**Summary Of The Review:**

This paper presents a fact that using combined audio features with facial feature could improve facial animation.
However, there is not any novel design or insight about this statement.
Using StyleGAN2 to generate image is not a contribution.
StyleGAN2 might not be applicable for video generation due to lack of speech details.

---

### Official Review · Reviewer_FwS3 · 2022-10-25

**Confidence:** 5
**Clarity, Quality, Novelty And Reproducibility:** 1. The clarity and quality in terms o…
**Correctness:** 2
**Technical Novelty And Significance:** 1
**Empirical Novelty And Significance:** 1
**Recommendation:** 1

**Details Of Ethics Concerns:**

None.

**Strength And Weaknesses:**

Strength:
1. The reason for using StyleGAN2 as the generator is discussed with an illustration.

Weaknesses:
1. The language fails to convey the meaning. Please revise the paper with the help of English professionals.
2. The cited state-of-the-arts (SOTAs) are out-of-date. Please cite and make a comparison with more recent papers, such as [1][2][3][4].
3. The method is proposed unclearly. For example, why are there two StyleGAN2 in Figure 2? Moreover, the Decoder mentioned in section 3.1 is missing in Figure 2.
4. The novelty of the proposed method has not been discussed in the paper.
5. The generated results should be compared with SOTAs visually.
6. Minor issues:
    a) Mcc Encoder in Figure 2 should be MFCC Encoder.
    b) The second line in section 3.4, "... as shown in the figure appears" should be Figure 3.
    c) To be formal, I suggest the author use "adversarial" rather than "antagonistic" and use "concatenate" rather than "splice".

[1] Wu, Haozhe, et al. "Imitating arbitrary talking style for realistic audio-driven talking face synthesis." Proceedings of the 29th ACM International Conference on Multimedia. 2021.
[2] Zhou, Hang, et al. "Pose-controllable talking face generation by implicitly modularized audio-visual representation." Proceedings of the IEEE/CVF conference on computer vision and pattern recognition. 2021.
[3] Ye, Zipeng, et al. "Audio-driven talking face video generation with dynamic convolution kernels." IEEE Transactions on Multimedia (2022).
[4] Ji, Xinya, et al. "EAMM: One-Shot Emotional Talking Face via Audio-Based Emotion-Aware Motion Model." SIGGRAPH, 2022.

**Summary Of The Paper:**

This paper proposed to use StyleGAN2 as the generator to synthesize talking head videos, taking as input the ID feature and the video or audio feature. The consistency of video and audio features is constrained by an adversarial loss. The experiments are conducted on the LRW and CREAM-D datasets.

**Summary Of The Review:**

For the above reasons, this paper is entirely below the acceptance threshold.

---

### Official Review · Reviewer_MS16 · 2022-10-27

**Confidence:** 4
**Correctness:** 2
**Technical Novelty And Significance:** 2
**Empirical Novelty And Significance:** 2
**Recommendation:** 3

**Clarity, Quality, Novelty And Reproducibility:**

Overall, the paper proposes to combine existing techniques into a StyleGan2-like generator to solve the problem of video generation, conditioned on target identity, audio signal (which should drive the mouth expressions) or video signal.

I found the proposed method being only an incremental extension to existing techniques (Bulat et al. 2017, Zhou et al. 2019 etc), whereas an important discussion on details/modifications have been neglected (e.g. in sec. 3.5 there is no information on how the discriminator is used for optimising the sequence).

The discussion in the evaluation section is quite limited without proper explanations of metrics and what's the impact of different proposed components (e.g. the impact of the synchronisation discriminator is not measured with any of the metrics, only a few pictures are provided)

Finally, the paper requires proof reading. It contains multiple English grammar errors and uses non-scientific language. Examples:
“Fool the discriminator maximally”
“Our approach is feasible” - missing proper context
“ generation from speech began to flourish”
Figure 2 is referred as Figure 3 (Section 3, the first sentence)
Different sections introduce variables that are never referred to, or are referred after multiple sections (see. 3.3. Audio encoder, E_a)
The legend in Fig 6. has too small font size.
We spliced the features -> combined?

**Strength And Weaknesses:**

Pros:
- An interesting approach for generating talking faces
Cons:
- The paper requires proof reading
- Limited novelty: most of the components of the proposed solution have already been published
- Limited evaluation and ablation analysis




**Summary Of The Paper:**

This paper addresses the problem of generating a talking face animation video driven by audio (or video), conditioned on a target face appearance. The proposed approach consists of a StyleGan2-like generator conditioned on output from identity, video and audio encoders.

**Summary Of The Review:**

Recommend to Reject.

---

### Decision · Program_Chairs · 2023-01-20

**Decision:**

Reject

**Justification For Why Not Higher Score:**

Although the proposed idea is interesting and well-motivated, all four reviewers were in agreement that this paper is not ready for publication at the current stage due to limited novelty, experiments and missing details. Further, there was no rebuttal, and the reviewers' opinions did not change after the rebuttal period. Please keep the reviewers' comments in mind when preparing a future version of the manuscript.

**Justification For Why Not Lower Score:**

N/A

**Metareview: Summary, Strengths And Weaknesses:**

This paper presents a method for audio-driven talking face generation conditioned on target face image/video. The architecture uses three separate encoders for extracting the acoustic features, the lip motion, and the identity. These learned representations are then concatenated and used as input to StyleGAN2 models for generating the output sequence. The experiments are conducted on the LRW and CREAM-D datasets.

Strengths:
1. The addressed problem has real-world applications.
2. The idea of combining audio and image features for generation task  is interesting and well-motivated.

Weaknesses:
1. The idea has limited novelty -- most of the components of the proposed solution have already been published and the method is an incremental extension to existing techniques (Bulat et al. 2017, Zhou et al. 2019 etc).
2. The evaluation is limited with missing ablation.
3. The cited SOTAs are out-of-date.
4. There are missing details for reproduction, such as the detailed structures of the networks, and how the discriminator is used for optimizing the sequence.
5. The clarity and quality in terms of language and logic are not good. It contains multiple English grammar errors and requires proof reading.